# Detection of retinal changes with optical coherence tomography angiography in mild cognitive impairment and Alzheimer's disease patients: A meta-analysis

Jingwen Hui[1,2,3,4&], Yun Zhao[1,2,3,4&], Shasha Yu[1,2,3,4], Jinfeng Liu[5], Kin Chiu[5,6]*, Yan Wang[1,2,3,4]*

**1** Tianjin Eye Hospital, Tianjin Eye Institute, Heping District, Tianjin, China, **2** Tianjin Key Lab of Ophthalmology and Visual Science, Tianjin, China, **3** Clinical College of Ophthalmology Tianjin Medical University, Tianjin, China, **4** Nankai University Eye Hospital, Tianjin, China, **5** Department of Ophthalmology, Li Ka Shing Faculty of Medicine, The University of Hong Kong, Hong Kong SAR, China, **6** State Key Laboratory of Brain and Cognitive Sciences, The University of Hong Kong, Hong Kong SAR, China

& These authors contributed equally to this work.
* datwai@hku.hk (KC); wangyan7143@vip.sina.com (YW)

**Data Availability Statement:** All files are available from PubMed (https://pubmed.ncbi.nlm.nih.gov/) and Embase (https://www.embase.com/landing?

## Abstract

### Objective

To assess retinal microvascular network impairments in the eyes of mild cognitive impairment (MCI) and Alzheimer's disease (AD) patients with optical coherence tomography angiography (OCTA).

### Design

Systematic review and meta-analysis.

### Methods

A literature search was conducted in the PubMed and EMBASE databases to identify relevant studies detecting retinal microvascular attenuation among AD, MCI patients and cognitively healthy controls (HCs) by OCTA. Data were extracted by Review Manager V.5.4 and Stata V.14.0.

### Results

Eight investigations were included in this meta-analysis, with 150 AD patients, 195 MCI patients and 226 HCs were eligible for meta-analysis. Evidence based on these studies demonstrated that there was a significantly decreased vessel density (VD) of the Optovue group in superficial capillary plexus (SCP): WMD = -2.26, 95% CI: -3.98 to -0.55, p = 0.01; in deep capillary plexus (DCP): WMD = -3.40, 95% CI: -5.99 to -0.81, p = 0.01, VD of the Zeiss group in SCP:WMD = -0.91, 95% CI: -1.79 to -0.02, p = 0.05 and an enlarged fovea avascular zone (FAZ):WMD = 0.06, 95% CI: 0.01 to 0.11, P = 0.02 in OCTA measurements of MCI patients. Additionally, in OCTA measurements of AD patients, there was a significantly

status=grey) following the protocol outlined in the article. Please see "Data sources and search strategy" and "Study selection and data extraction" in the Methods section for the search criteria used.

**Funding:** This study was supported by grant from the National Natural Science Foundation of China (Grant No:81873684).The funders had no role in study design, data collection and analysis, decision to publish, or preparation of the manuscript.

**Competing interests:** The authors have declared that no competing interests exist.

decreased VD in the SCP: WMD = -1.88, 95% CI: -2.7 to -1.07, p<0.00001. In contrast, there was no significant decrease in DCP nor enlargement of FAZ in AD patients.

## Conclusion

Retinal microvascular alternations could be optimally screened in MCI patients detected by OCTA, which could be a warning sign of relative changes in the MCI before progressing to AD. Retinal microvasculature changes worth further investigation in larger scale clinical trials.

## Introduction

Alzheimer's disease (AD) is progressive neurodegenerative disease, and the most common cause of dementia with cognitive impairment in the elderly. An estimated 40 million people worldwide have AD, and due to extending lifespans, this number is only expected to increase [1]. While in the preclinical stage mild cognitive impairment (MCI), the cognitive function of patient declines surpasses the expectation when taking the age into consideration. Because of no impaired performance of daily life activities, it is a transitional period of time in which the crucial pathophysiologic changes of AD exist. However, the critical symptoms cannot be detected clearly [2]. Once AD is diagnosed, patient will have an irreversible process, leading to dementia starts. On average patient can live for 4.6 years longer once diagnosed [3]. MCI possibly is a suggestion of preclinical stage of AD. Therefore, early diagnosis of the preclinical stage is important before the irreversible process begins is important. High-risk target population for progression to AD are those who obtained the amnestic type of MCI [4, 5].

Early diagnostic efficient testing of AD and MCI is still challenging and difficult [6]. Nowadays, we rely on clinical evaluation to diagnose MCI, such as neuropsychological testing [7]. Neuroimaing findings of the brain and spinal-fluid examination can offer objective analysis to correlate with patient's cognitive changes. However, these methods are extremely expensive and invasive [8]. Therefore, the discovery of new noninvasive and economical screening tool for diagnosis of MCI and AD is a major goal of in current research and numerous people have dedicated into this field [9].

Due to the shared dicencephalic origin of the retina and brain, retina is regarded as an extension of the central nervous system (CNS) [10, 11]. There is homology between retinal and cerebral vasculature. Optical coherence tomography (OCT) is a technique that allows imaging of the retina with a micrometer resolution [12]. The advanced functional extension of OCT termed OCT- angiography (OCTA) [13] can help a detailed angiographic view of the retinal vascular network [14, 15].

The loss in retinal vessel density and expansion of the FAZ area detected by OCTA could reflect changes in the MCI patients. OCTA scans and parameters have demonstrated the loss of the retinal vessel density is related with preclinical AD, possibly sharing a useful approach to neurodegenerative progress. Taking these limitations of the previous meta-analysis into the consideration, a review on OCTA system were performed and meta-analyzed to assess the changes among MCI, AD patients and healthy controls (HCs). Studies with detailed examine parameters of MCI, AD and HC were included.

## Methods

The meta-analysis was carried out in accordance with the guidance illustrated by the Meta-Analysis of Observational Studies statement [16].

## Data sources and search strategy

Comprehensive literature searching was conducted by two independent reviewers (HJW and ZY). Relevant studies were identified by searching PubMed and Excerpta Medica Database (EMBASE) according to the following criteria:(Alzheimer's disease [Title/Abstract] OR mild cognitive impairment [Title/Abstract] OR "AD[Title/Abstract]"mild Alzhermer's disease) AND ("optical coherence tomography angiography"[Title/Abstract] OR "OCT angiography" [Title/Abstract]) OR OCTA[Title/Abstract] OR "angio-OCT"[Title/Abstract]. Clinical studies published in peer-reviewed journals before March 31st 2021 were included. The gray literature and unpublished data were also measured. The literatures were analyzed with the bibliographic database.

## Study selection and data extraction

The inclusion criteria:

1. Studied of MCI and AD patients or included the preclinical stage of AD

2. Measurements recorded as the mean and standard deviation (SD)

3. Observational comparative studies

4. Research work in English

   The exclusion criteria:

1. Studies that used for analysis without measure the methods in subjects with AD, MCI, HC.

2. Conference abstracts, letters to editor, non-English records, animal studies and case reports.

3. Insufficient data to estimate a weighted mean difference (WMD).

4. Duplicate study populations or redundant publications.

## Quality assessment

Quality evaluation was independently conducted by three reviewers (HJW, ZY and YSS). The other two reviewers searched and contacted relevant research when the information were inaccessible (JFL and KC). After the duplicate literatures were removed, the titles and abstracts of the remained literatures were screened according to the inclusion criteria and the exclusion criteria. The following data were extracted from each study: the first author, year of publication, design, country or region, sample size, type of AD, mean age, OCTA device used and details of OCTA scans. We chose the larger scan for a more satisfying resolution when results were available in various sizes of scans in the same study.

## Statistical analysis

Review Manager V.5.4 (Cochrane Collaboration, Oxford, United Kingdom) and Stata V.14.0 were used for statistical analysis. In the analysis, continuous variables extracted is the mean values and SDs can be figured out using WMDs and the 95% confidence intervals (CIs). The Chi-square test and $I^2$ statistic were applied to assessing the heterogeneity in statistic, and the use of the random-effect model was for predicting the levels of heterogeneity. Publication bias was evaluated by Egger's linear regression test and $P<0.05$ regarded as statistically significant.

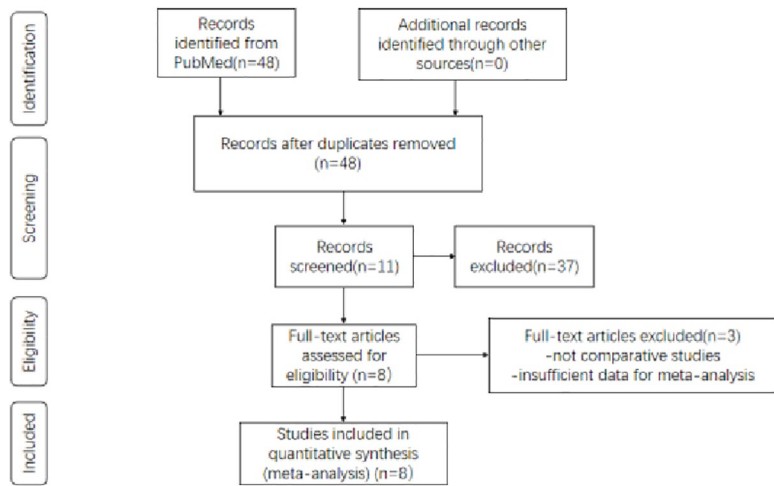

**Fig 1. The flowchart of the research selection.** A total of eight potentially relevant articles were identified from our search strategies across all databases and systematic review reference lists.

## Results

Original research work published between January 1st 2015 and March 31st 2021 were covered, the literature search yielded 48 results (Fig 1), of which 37 were removed after a title and abstract screen by inclusion/exclusion criteria. Another three studies were excluded because of lack of sufficient data for analysis after full-text checking of the remaining 11 studies. Eight cross-sectional OCTA studies on the retinal changes among AD, OCTAMCI patients, and HC were included in this meta-analysis [2, 6, 17–20, 30, 31]. Studies published between January 1st 2015 and March 31st 2021 were covered. In total, 150 AD patients, 195 MCI patients and 226 HCs were included in this meta-analysis (Table 1).

Overall, from these 8 studies on the mean vessel density (VD) in superficial capillary plexus (SCP) layer, 7 studies suitable for analysis were found including 113 AD (approximately 68 female) patients, 195 MCI (100 female) patients, and 197 healthy controls (111 female). For the mean VD in the deep capillary plexus (DCP) layer, we identified 4 studies suitable for analysis, which included 104 AD (61 female) patients, 132 MCI (69 female) patients, and 128 healthy controls (70 female). For the fovea avascular zone (FAZ) area, 4 studies were figured out suitable for the evaluation, this included 89 AD (49 female) patients, 104 MCI (56 female) patients, and 115 healthy controls (65 female). Table 1 presents the characteristics of all articles, and Figs 2 and 3 showed the main results of the meta-analysis.

Due to the differences in OCTA technology and site for the segmentation of retinal layers among various types of devices, VD results in the studies were not compared among studies that used different machines. Moreover, VD used in different machine is either area-based measurements (Optovue), length-based measurements (Zeiss). Our analysis focused on the outcomes using the same machine. There were 4 studies used the Optovue OCTA machine to analysis VD in the SCP (Table 2, Fig 2A) between MCI patients and HC healthy controls. Because of the obvious heterogeneity ($I^2 = 70\%$), the random effects model was adopted to pool the data. The meta-analysis indicated that the average VD in MCI patients was declined obviously when comparing with that in health ones (WMD = -2.26, 95% CI: -3.98 to -0.55, p = 0.01, Fig 2A). The analysis of mean vessel density (length based) in the SCP layer in 3 Zeiss studies (Table 2, Fig 2B) between MCI and HC found significant heterogeneity ($I^2 = 64\%$) over the studies, so the data were pooled through the random effects model. The meta-analysis of

**Table 1. Included studies using OCTA to assess the retinal microvasculature in Alzheimer's disease (AD), mild cognitive impairment (MCI) patients and healthy controls (HC).**

| Reference | No. of subjects and diagnosis | Age (years) | Gender (female/male) | Mean VD in SCP | Mean VD in DCP | Mean FAZ (mm2) |
|---|---|---|---|---|---|---|
| Chua 2020 [17] | 37 MCI | 77.9±6.4 | 16/21 | 14.94±1.02 | 20.81±1.65 | - |
| | 24 AD | 74.9±6.0 | 17/7 | 14.78±1.14 | 20.42±1.60 | - |
| | 29 HC | 76.7±5.3 | 13/16 | 15.66±0.96 | 21.54±1.55 | - |
| Criscuolo 2020 [5] | 27 MCI | 73.0±6.0 | 15/12 | 44.92±5.04 | 45.13±6.67 | 0.28±0.12 |
| | 29 HC | 73.1±7.0 | 15/14 | 48.12±4.53 | 50.58±4.69 | 0.19±0.06 |
| Shin 2021 [18] | 40 MCI | 72.8±8.6 | 15/25 | 14.0±3.9 | 25.5±1.9 | - |
| | 37 HC | 69.0±10.4 | 20/17 | 16.3±2.5 | 25.6±1.8 | - |
| Wang 2021 [19] | 47 MCI | 72.73±7.75 | 29/18 | 44±3.07 | 49.57±2.89 | 0.36±0.12 |
| | 62 AD | 71.81±7.98 | 35/27 | 44.66±3.36 | 49.42±3.4 | 0.34±0.11 |
| | 49 HC | 69.5±5.94 | 32/17 | 46.82±2.08 | 50.89±2.86 | 0.33±0.12 |
| Wu 2020 [2] | 21 MCI | 67.81±5.96 | 9/12 | 50.37±2.33 | 48.09±3.88 | 0.37±0.06 |
| | 18 AD | 69.94±6.39 | 9/10 | 49.56±2.81 | 43.10±2.75 | 0.44±0.08 |
| | 21 HC | 68.67±5.85 | 10/11 | 50.47±2.72 | 52.28±2.89 | 0.26±0.07 |
| Yan 2021 [20] | 37 AD | - | - | 15.8±6.975 | 28.8±8.147 | - |
| | 29 HC | - | - | 15.94±6.264 | 28.8±8.298 | - |
| Yoon 2019 [30] | 7 MCI | 70.70±9.10 | 3/4 | 17.77±1.18 | - | 0.17±0.05 |
| | 9 AD | 75.20±7.50 | 5/4 | 18.31±0.52 | - | 0.17±0.10 |
| | 16 HC | 73.30±8.30 | 8/8 | 17.98±0.99 | - | 0.17±0.07 |
| Zhang 2019 [31] | 16 MCI | 73.03±8.24 | 13/3 | 40.67±5.23 | - | - |
| | 16 HC | 73.60±7.69 | 13/3 | 44.50±4.11 | - | - |

SCP: superficial capillary plexus; DCP: deep capillary plexus; (-): no enough data recorded in the article.

these data showed that the mean VD in MCI patients was decreased significantly when comparing with that in healthy group (WMD = -0.91, 95% CI: -1.79 to -0.02, p = 0.05, Fig 2B). The meta-analysis of mean VD in DCP data showed that there was a significant heterogeneity ($I^2$ = 80%) over 3 studies, using Optovue, so the random effects model pooled the data. The meta-analysis on that indicated that the average VD in DCP of MCI patients was declined significantly when comparing with that in the HC (WMD = -3.40, 95% CI: -5.99 to -0.81, p = 0.01, Fig 2C). Measurement of area size in the FAZ can be compared among different machines. In this part of the meta-analysis, we pooled 4 studies together using the random effects model due to the high heterogeneity ($I^2$ = 79%). The data of the meta-analysis presented that the mean vessel density in the FAZ area of MCI patients was increased significantly compared with HC (WMD = 0.06, 95% CI: 0.01 to 0.11, p = 0.02, Fig 2D). Therefore, when compare MCI patients with healthy control, there was significant VD decrease in both SCP and DCP, together with enlargement of FAZ area.

In the 3 studies that Optovue OCTA machine used to compare the analysis of the mean VD (area-based) in the SCP layer between AD and HC figured out significance of heterogeneity ($I^2$ = 0%) over the studies, so we pooled the data via fixed effects model. The meta-analysis of these data showed that the mean VD in AD patients was reduced significantly compared with that in the HC (WMD = -1.88, 95% CI: -2.7 to -1.07, p<0.00001, Fig 3A). However, the further analysis of VD in the DCP, and FAZ area there was no statistical difference (P>0.05, Fig 3B & 3C) between AD patients and HC. To sum up, the outcomes presented that a significance of vessel density decrease in the SCP of people with AD comparing with the healthy controls. To sum up, the outcomes presented that a significance of vessel density decrease in the SCP of people with AD comparing with the healthy controls.

### A. OPTOVUE Vessel density in SCP

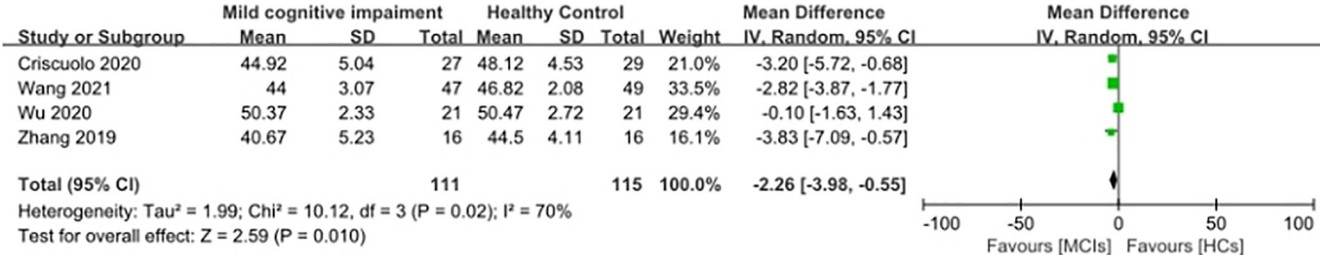

### B. ZEISS Vessel density in SCP

### C. OPTOVUE Vessel density in DCP

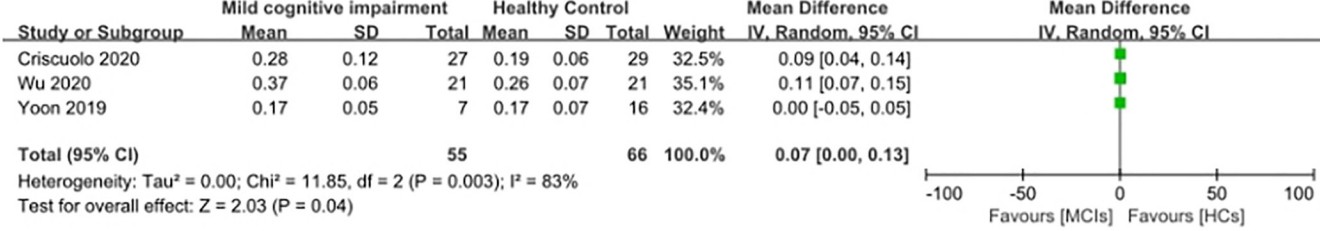

### C. FAZ area

**Fig 2. Meta-analysis of vessel density (VD) for mild cognitive impairment patients versus healthy controls.** Mean and standard deviation (SD) are included, with 95% confidence intervals (CIs), heterogeneity scores and overall effect in an inverse variance (IV) random effects model. The green square size represents the weight attributed to each study based on relative sample size. (A) VD in superficial capillary plexus (SCP) detected by Optovue. (B) VD in SCP detected by Zeiss. (C) VD in deep capillary plexus (DCP) detected by Optovue. (D) Area size of foveal avascular zone (FAZ) in mm2.

## Discussion

This meta analysis focused on retinal microvascular changes in AD, MCI patients and HC. Comparing with HC, there was decreased VD in both SCP and DCP together with enlarged FAZ area in the MCI patients. In the studies included in this analysis, there was decreased VD in SCP in AD patient.

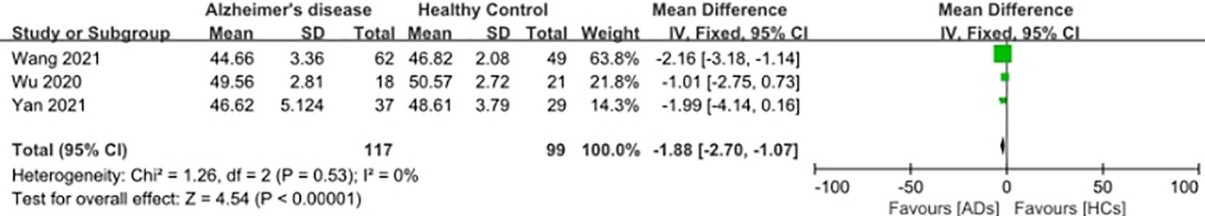

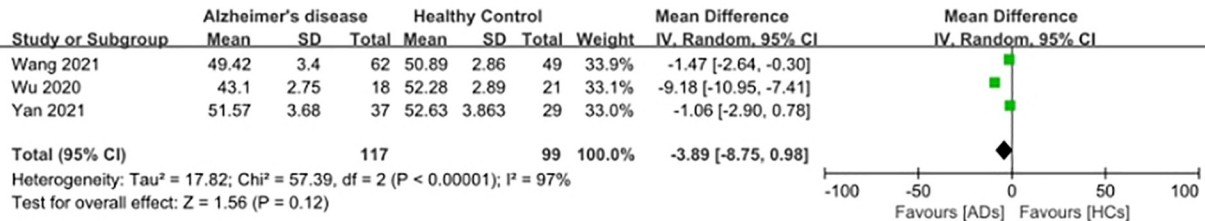

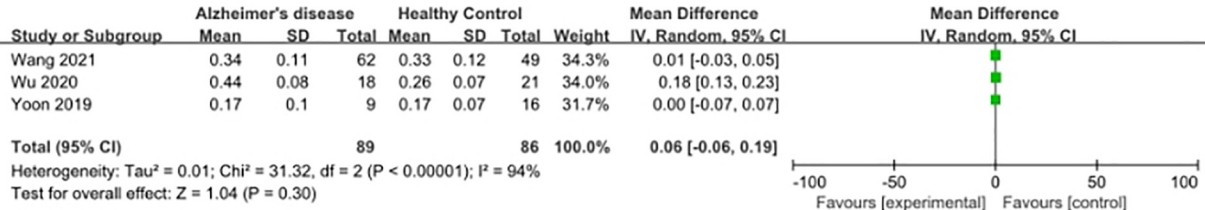

**Fig 3. Meta-analysis of vessel density (VD) for Alzheimer's disease patients versus healthy controls.** Mean and standard deviation (SD) are included, with 95% confidence intervals (CIs), heterogeneity scores and overall effect in an inverse variance (IV) random effects model. The green square size represents the weight attributed to each study based on relative sample size. (A) VD in superficial capillary plexus (SCP) detected by Optovue. (B) VD in deep capillary plexus (DCP) detected by Optovue. (C) Area size of foveal avascular zone (FAZ) in mm2.

**Table 2. Measurements that were included 8 studies are shown, each study was identified by using either the Optovue or the Zeiss OCTA machines.**

| Study | OCTA machine | Superficial VD | Deep VD | FAZ area |
|---|---|---|---|---|
| Criscuolo et al. | Optovue | ↓MCI | ↓MCI | ↑MCI |
| Wang et al. | Optovue | ↓MCI↓AD | ↓MCI↓AD | ~MCI~AD |
| Wu et al. | Optovue | ~AD~MCI | ↓AD↓MCI | ↑AD↑MCI |
| Yan et al. | Optovue | ↓AD | ~AD | - |
| Zhang et al. | Optovue | - | ~MCI/AD | ~MCI/AD |
| Chua et al. | Zeiss | ↓MCI↓AD | ↓AD~MCI | - |
| Yoon et al. | Zeiss | ↓MCI~AD | ↓MCI~AD | ~MCI~AD |
| Shin et al. | Zeiss | ↓MCI | ↓MCI | ↑MCI |

(↑): indicates evidence of a significant increase in the case group compared to the control group. (↓): indicates a significant decrease, (-): not enough data, and (~): indicates no evidence of a significant difference.

There are two major issues to be considered when we do meta-analysis on OCTA images. First is the scan size of the image. In the current study, 6×6 mm$^2$ OCTA images were analyzed. Although this may be more affected by movement artifacts, it can capture a larger macular area, which means that more retinal vessels are captured for better evaluation. For the studies included in the current analysis, the quality of the OCTA data was reliable. Second is the various machines for performing measurements. Each device uses specific algorithms to reconstruct the images and gives different terminology to describe the outcome. The Optovue uses AngioVue software which operates using a split-spectrum amplitude decorrelation angiography algorithm (SSADA). The Zeiss uses AngioPlex software which uses an optical microangiography (OMAG) approach. Topcon employs OCTA ratio analysis (OCTARA). Our study compared those results from different reports using same machine, the conclusion that MCI patients have decreased VD in both SCP and DCP was supported by both Optovue and Zeiss. Our study showed retinal capillary loss in the superficial and deep layers and an increased FAZ area in the retina on OCTA, manifesting the decrease of retinal microvascular density and impairment in MCI patients when comparing to the cognitively healthy controls. The changes our study identified supported the hypothesis that vascular impairment leading to hypoperfusion may contribute to the onset and progression of AD [21].

AD is known as a disease that mainly influence the brain and recent studies have shown that the retina could affected to some degree [22]. The retina is regarded as the window into the brain, of which the microvasculature may yield insightful information into the pathophysiology of early AD [9, 23]. Due to the homology between retinal and cerebral vasculature, retinal vascular network changes in MCI could be measured using OCTA. OCTA is a rapid and efficient imaging and testing approach of alternations in the retina correlated to MCI, whose progression shows the evolution of AD [24]. The loss in retinal vessel density and expansion of the FAZ area detected by OCTA could reflect changes in the MCI population [25, 26]. OCTA scans and parameters have demonstrated the loss of the retinal vessel density is related with preclinical AD, possibly sharing a useful approach to neurodegenerative progress [27]. Timely diagnosis of MCI plays a leading role in the evaluation of disease progression and treatment efficacy [28]. This is the first meta-analysis which covers all accessible high-quality studies and considers OCTA to assess microvascular impairment in the eyes of MCI and AD patients when taking cognitively healthy controls into consideration.

Most of these previous studies were conducted in AD and few used in MCI patients. While VD was reported decline in AD patient, those in MCI were limited and conflicting. The FAZ area difference between people with MCI patients and HC was unclear [28]. Thus, we summarized the related studies and found that reduced VD and enlarged FAZ area may highlight the role of retinal microvasculature in the MCI process rather than in the late stage when AD is diagnosed. MCI patients had significant enlargement of the FAZ area and notably decreased VD of the macula in the superficial and deep retinal layers. We analyzed the VDs of the SCP and DCP, which were the percentage of the peripapillary region occupied by superficial and deep capillaries, respectively [29]. Furthermore, we found a significant difference in the FAZ area between MCI patients and HCs, which suggests that the FAZ could be another useful indicator in the early stage of preclinical AD. However, the indication about the changes in VD of the DCP and FAZ area measurements are still controversial. Decreased VD especially in DCP at the stage of MCI might be useful to elucidate the mechanism of retinal capillary pathology in MCI. For instance, Yoon et al. [30] and Zhang et al. [31] showed a critically lower VD in the whole retina but not in the DCP layer, and no differences in the FAZ area were observed between them. Our meta-analysis found VD not only decreased in SCP but also DCP in MCI patients with significant FAZ area enlarged. This indicated the low perfusion status in the MCI patients in the retina and require further clinical investigation.

The meta-analysis also proved there was a significant VD reduction in the SCP of AD patients compared with HC. And no significant change in VD in DCP layer and FAZ area. Three studies using Optovue offered measurements of VD for the SCP and DCP and FAZ area between the AD patients and HC. Wang et al. (2021) [19] and Yan et al. (2021) [20] did not find evidence of a difference in SCP, DCP and FAZ area between AD and HCs. Wu et al. (2020) [2] found a change in DCP and FAZ but no change in SCP between AD patients and HCs. Comparing to HC, the significant decrease of VD in the SCP was found in MCI and AD patients. This is consistent with the reports that inner retinal thickness of AD patients detected by OCT, reducing retinal microvasculature correlated with loss of inner retinal neurons. VD decrease in DCP and enlargement of FAZ area in the MCI stage might be transient and compensated in the progression of AD.

There are some limitations of this meta-analysis: 1) Even if the sensitivity analysis was carried out with great care, the heterogeneities of some comparisons still important, possibly relating to factors including the use of manual measurements and the inclusion of different ethnicities. 2) Because of the insufficiency of published findings, some possibly influence on measuring the OCTA, like axial length which were not studied. If future studies had increased samples sizes, it would be more helpful to support the OCTA screening application of early MCI. 3) The process of progressing from a relatively healthy condition to the mild cognitive impairment stage may be long. 4) What need to be mentioned is that the metrics in the work here contain inherent limits which could lead to the inconsistent outcomes viewed while in the literature. So the meta-analysis was insufficient for the exploration on the specific timing and process of microvascular impairments according to the cross-sectional studies covered, meaning that we need more well-designed longitudinal studies.

To sum up, there is a significantly decreased VD and enlarged FAZ area in OCTA measurements between people with MCI and HC. The data indicate that retinal microvascular alterations could be optimally screened in MCI patients using OCTA, which could warn of relative changes in the brain before progressing to AD. There was a significantly decreased VD in SCP in OCTA measurements between AD patients and HC However, there was no significant change in VD in the DCP and FAZ area between AD patients and HC, which revealed that retinal microvascular changes could persistent in superficial layers than in deep layers. OCTA detection of the VD in the SCP might be a useful biomarker for MCI and AD diagnosis. In the future, investigations with a larger sample size are needed to assess the effectiveness of this method and verify the association of retinal microvasculature changes in MCI, AD and HCs.

## Supporting information

**S1 Checklist.**
(DOC)

**S1 File. Newcastle-ottawa quality assessment scale case control studies.**
(DOCX)

## Author Contributions

**Data curation:** Jingwen Hui, Shasha Yu, Jinfeng Liu, Kin Chiu.

**Formal analysis:** Jingwen Hui, Shasha Yu, Jinfeng Liu, Kin Chiu.

**Funding acquisition:** Yan Wang.

**Investigation:** Yun Zhao.

**Methodology:** Yun Zhao.

**Project administration:** Yun Zhao.

**Resources:** Jingwen Hui, Yun Zhao, Kin Chiu, Yan Wang.

**Validation:** Jingwen Hui, Kin Chiu, Yan Wang.

**Visualization:** Kin Chiu.

**Writing – original draft:** Jingwen Hui.

**Writing – review & editing:** Jingwen Hui, Shasha Yu.

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
