## [Decision Letter · Decision Letter 0]

11 May 2021

PONE-D-21-08825

Detection of retinal changes with optical coherence tomography angiography in mild cognitive impairment and Alzheimer’s disease patients: a meta-analysis

PLOS ONE

Dear Dr. Wang,

Thank you for submitting your manuscript to PLOS ONE. After careful consideration, we feel that it has merit but does not fully meet PLOS ONE’s publication criteria as it currently stands. Therefore, we invite you to submit a revised version of the manuscript that addresses the points raised during the review process.

Whilst of interest, the reviewers have raised major concerns on the methodology and criteria of the meta-analysis.

We look forward to receiving your revised manuscript.

Kind regards,

Rayaz A Malik, MBChB, PhD

Academic Editor

PLOS ONE

Journal Requirements:

PLOS ONE does not copy edit accepted manuscripts (https://journals.plos.org/plosone/s/criteria-for-publication#loc-5). To that effect, please ensure that your submission is free of typos and grammatical errors.

5. Thank you for submitting the above manuscript to PLOS ONE. During our internal evaluation of the manuscript, we found significant text overlap between your submission and the following previously published work:

- https://journals.plos.org/plosone/article?id=10.1371%2Fjournal.pone.0134750

Please revise the manuscript to rephrase the duplicated text, cite your sources, and provide details as to how the current manuscript advances on previous work. Please note that further consideration is dependent on the submission of a manuscript that addresses these concerns about the overlap in text with published work.

Additional Editor Comments:

Both reviewers find the meta-analysis of interest and relevance.

However, they highlight major areas which need to be addressed.

Reviewers' comments:

Reviewer's Responses to Questions

**Comments to the Author**

1. Is the manuscript technically sound, and do the data support the conclusions?

Reviewer #1: Partly

Reviewer #2: No

2. Has the statistical analysis been performed appropriately and rigorously? 

Reviewer #1: Yes

Reviewer #2: Yes

3. Have the authors made all data underlying the findings in their manuscript fully available?

Reviewer #1: Yes

Reviewer #2: Yes

4. Is the manuscript presented in an intelligible fashion and written in standard English?

Reviewer #1: No

Reviewer #2: No

5. Review Comments to the Author

Reviewer #1: This meta-analysis article shows that the retinal vessel density (RVD) in patients with MCI was significantly lower in superficial (SCP) and deep capillary plexus (DCP) and higher in fovea avascular zone (FAZ) compared to healthy controls (HC). However, there was no significant difference in RVD in these regions between patients with Alzheimer’s disease (AD) and HCs.

This paper requires major revision and English editing.

The conclusion that the authors draw from their meta-analysis that RVD could screened in MCI to predict AD is invalid since RVD is not associated with AD. They should provide a possible explanation on why there is no difference in RVD between patients with AD and HC. That of what is written in the 3rd paragraph in the discussion is not an explanation.

The authors should include vascular dementia (VaD) and mixed dementia separately in their meta-analysis. They might find an association of RVD with VaD and mixed dementia.

Need to define all abbreviations in the abstract. Need to define vessel density and its abbreviation in the introduction.

The statement that only 5 cross-sectional studies were included in the meta-analysis in the first paragraph in the results is inconsistent with the other results (8 studies for RVD in SCP etc.) and abstract.

Deeming the cognitive assessments for MCI diagnosis as “time-consuming” in the introduction is inappropriate. Repeating a battery of cognitive assessments is essential for diagnosis of MCI.

Correct “vessel length density” in the last sentence in the third paragraph in the results.

Add references in the discussion. Which studies showed decreased VD in AD? Need to discuss which studies were conflicting on the association of RVD with MCI?

The RVD measurements in the studies from Chua, Shin, Yan and Yoon are very low compared to the other studies. Please provide a reason for the differences (i.e. manual measurement methods) and mention specifically about the studies.

Reviewer #2: Manuscript Title: Detection of retinal changes with optical coherence tomography angiography in mild cognitive impairment and Alzheimer’s disease patients: a meta-analysis

Summary: This systematic review assesses differences in OCT-A vascular density measurements within superficial and deep retinal capillary vessels amongst patients diagnosed with Alzheimer’s disease or mild cognitive impairment compared to health controls. The study topic is of importance as non-invasive methods for detecting early cognitive decline would be significantly beneficial for screening and directing preventative interventions. However, several methodological issues in the conduction of the systematic review mean the conclusions are not supported by evidence as presented in the manuscript.

Major Issues:

1. The authors should elaborate further on the following statement: “Take these limitations of the previous meta-analysis into the consideration”. It is unclear which prior meta-analysis they are referring to as this is not discussed/referenced in the introduction.

a. There are a number of reviews assessing the association between OCT and AD, most recently by Chan et al 2019.

2. Further detail is needed in the inclusion criteria, in particular, what are the definitions of MCI, pre-clinical AD and AD used to select studies for the systematic review.

a. There authors should consider extracting data that would help quantify the level of MCI/AD in the included studies (e.g. MMSE score).

3. The authors should provide a detailed search strategy with the N for results for the given search terms and databases searched.

4. The attached risk of bias assessment appears to be for randomised-controlled trials and not for diagnostic studies. Furthermore, the risk of bias assessment is not clearly described in the manuscript as there is only the mention of a ‘Quality evaluation’.

5. A significant issue in comparing OCT-A vessel density measurements is the fact that measurements vary significantly between OCT-A devices and extraction techniques. Many VD extraction techniques are proprietary, therefore, it can be challenging to compare study measurements. Was the OCT-A device and extraction technique considered factors during the conduction of the systematic review/meta-analysis?

a. In Table 1, there are large differences in the VD measurements between the studies with some reporting SCP VD of ~14-16% and other studies ~40-50%.

b. This could affect the sizes of mean differences between sub-groups and potentially studies should be grouped according to OCT-A device and not-pooled together as they are currently.

6. The 6*6mm macula scans were chosen over 3*3mm scans by default but I would question whether they have a better ‘resolution’. Although the captured macula area is bigger for the 6*6mm which would capture more of retinal vessels, this can come at the cost of more artefacts (a 6*6mm may be more affected by movement artifacts). The authors could instead extract data from both macula size scans and perform a subgroup analysis on the basis of the imaged macula area (6mm2 vs 3mm2).

7. Data on confounders of OCT-A vessel density should be recorded, where available, e.g. diabetes status, hypertensive status & retinal co-pathology.

8. The finding that MCI is associated with retinal microvascular changes but not AD (and that OCTA-should be used for MCI screening) is not sufficiently supported by the evidence-base as presented in the manuscript. The heterogeneity in all the analyses was high (I2>75%), hence, the certainly that can be placed on the summary estimates is limited.

9. Erratum? – Please clarify and amend the errors below.

a. The footnote for Fig 1 describes “A total of 4 potentially relevant articles were identified from our search strategies across all databases and systematic review reference lists” but the figure shows 9 studies in the meta-analysis.

b. Later in the results it details “After removing of duplicates and screening of titles and abstracts, 7 studies remained, and the full text were reassessed” but figure 1 shows 9 full-text articles were reviewed.

c. There after it mentions that “37 articles were excluded due to duplicated” but the figure suggests there were 48 articles after the removal of duplicates. The reasons for exclusion for the 37 studies is not given in Figure 1 or supplementary data.

10. The manuscript would benefit from scientific editing services as it requires refinement to improve its readability.

Minor Issues:

• Was the protocol for the now submitted literature review and meta-analysis published with the a-prior research questions and methodology?

• Some acronyms in the abstract are not defined within the abstract.

• Spacing is inconsistent and sentences are adjoined in places.

• Table 1 SCP and DCP VD units (% area) should be included.

6. PLOS authors have the option to publish the peer review history of their article (what does this mean?). If published, this will include your full peer review and any attached files.

Reviewer #1: **Yes: **Georgios Ponirakis

Reviewer #2: No

---

## [Author Response · Author response to Decision Letter 0]

4 Jul 2021

Major Issues:

1. The authors should elaborate further on the following statement: “Take these limitations of the previous meta-analysis into the consideration”. It is unclear which prior meta-analysis they are referring to as this is not discussed/referenced in the introduction.

a. There are a number of reviews assessing the association between OCT and AD, most recently by Chan et al 2019.

We have already revised and elaborated this part in the article.

2. Further detail is needed in the inclusion criteria, in particular, what are the definitions of MCI, pre-clinical AD and AD used to select studies for the systematic review.

a. There authors should consider extracting data that would help quantify the level of MCI/AD in the included studies (e.g. MMSE score).

We have added the definitions of MCI, pre-clinical AD and AD in the revised manuscript.

3. The authors should provide a detailed search strategy with the N for results for the given search terms and databases searched.

4. The attached risk of bias assessment appears to be for randomised-controlled trials and not for diagnostic studies. Furthermore, the risk of bias assessment is not clearly described in the manuscript as there is only the mention of a ‘Quality evaluation’.

We made a new NOS chart to assess the risk of bias between the studies which were included in our meta-analysis.

5. A significant issue in comparing OCT-A vessel density measurements is the fact that measurements vary significantly between OCT-A devices and extraction techniques. Many VD extraction techniques are proprietary, therefore, it can be challenging to compare study measurements. Was the OCT-A device and extraction technique considered factors during the conduction of the systematic review/meta-analysis?

a. In Table 1, there are large differences in the VD measurements between the studies with some reporting SCP VD of ~14-16% and other studies ~40-50%.

b. This could affect the sizes of mean differences between sub-groups and potentially studies should be grouped according to OCT-A device and not-pooled together as they are currently.

We had noted the measurements differences between studies. We only compared the results and pooled the data together between studies which used same machine.

6. The 6*6mm macula scans were chosen over 3*3mm scans by default but I would question whether they have a better ‘resolution’. Although the captured macula area is bigger for the 6*6mm which would capture more of retinal vessels, this can come at the cost of more artefacts (a 6*6mm may be more affected by movement artifacts). The authors could instead extract data from both macula size scans and perform a subgroup analysis on the basis of the imaged macula area (6mm2 vs 3mm2).

The 6*6mm scans captured more of retinal vessels in macula area than 3*3mm which means that the 3*3mm scans area was included by 6*6mm.

7.Data on confounders of OCT-A vessel density should be recorded, where available, e.g. diabetes status, hypertensive status & retinal co-pathology.

All the participants in our study were excluded by some complications like DM, hypertension and other retinopathy. 

8. The finding that MCI is associated with retinal microvascular changes but not AD (and that OCTA-should be used for MCI screening) is not sufficiently supported by the evidence-base as presented in the manuscript. The heterogeneity in all the analyses was high (I2>75%), hence, the certainly that can be placed on the summary estimates is limited.

In our study, we found that there was a significant difference of DCP and FAZ but no significant different of SCP between AD and HC. We concluded that retinal microvascular changes may occurred in different time.

9. Erratum? – Please clarify and amend the errors below.

a. The footnote for Fig 1 describes “A total of 4 potentially relevant articles were identified from our search strategies across all databases and systematic review reference lists” but the figure shows 9 studies in the meta-analysis.

b. Later in the results it details “After removing of duplicates and screening of titles and abstracts, 7 studies remained, and the full text were reassessed” but figure 1 shows 9 full-text articles were reviewed.

c. There after it mentions that “37 articles were excluded due to duplicated” but the figure suggests there were 48 articles after the removal of duplicates. The reasons for exclusion for the 37 studies is not given in Figure 1 or supplementary data.

We have already revised respectively in the article.

10. The manuscript would benefit from scientific editing services as it requires refinement to improve its readability.

The article has already been improved the fluency by scientific editing services.

Minor Issues:

• Was the protocol for the now submitted literature review and meta-analysis published with the a-prior research questions and methodology?

• Some acronyms in the abstract are not defined within the abstract.

• Spacing is inconsistent and sentences are adjoined in places.

• Table 1 SCP and DCP VD units (% area) should be included.

These issues were already solved in the article.

---

## [Editor Report · Decision Letter 1]

15 Jul 2021

Detection of retinal changes with optical coherence tomography angiography in mild cognitive impairment and Alzheimer’s disease patients: a meta-analysis

PONE-D-21-08825R1

Dear Dr. Wang,

We’re pleased to inform you that your manuscript has been judged scientifically suitable for publication and will be formally accepted for publication once it meets all outstanding technical requirements.

Kind regards,

Rayaz A Malik, MBChB, PhD

Academic Editor

PLOS ONE

Additional Editor Comments (optional):

The major concerns have been addressed. This analysis provides important insights on the use of OCT in MCI/dementia.
---

## [Editor Report · Acceptance letter]

27 Jul 2021

PONE-D-21-08825R1 

Detection of retinal changes with optical coherence tomography angiography in mild cognitive impairment and Alzheimer’s disease patients: a meta-analysis 

Dear Dr. Wang:

I'm pleased to inform you that your manuscript has been deemed suitable for publication in PLOS ONE. Congratulations! Your manuscript is now with our production department. 

Kind regards, 

on behalf of

Professor Rayaz A Malik 

Academic Editor

PLOS ONE